

# The novel role of LDHA/LDHB in the prognostic value and tumor-immune infiltration in clear cell renal cell carcinoma

Jie Chen[1], Fei Wu[2,3], Yehua Cao[4], Yuanxin Xing[5], Qingyong Liu[3] and Zuohui Zhao[6]

[1] Department of Urology, Jinan Central Hospital, Cheeloo College of Medicine, Shandong University, Jinan, Shandong, China

[2] Department of Urology, Shandong Provincial Hospital Affiliated to Shandong First Medical University, Shandong First Medical University & Shandong Academy of Medical Sciences, Jinan, Shandong, China

[3] Department of Urology, The First Affiliated Hospital of Shandong First Medical University & Shandong Provincial Qianfoshan Hospital, Jinan, Shandong, China

[4] Department of Gastroenterology, Jinan Central Hospital, Cheeloo College of Medicine, Shandong University, Jinan, Shandong, China

[5] Research Center of Basic Medicine, Jinan Central Hospital, Shandong First Medical University, Jinan, Shandong, China

[6] Department of Pediatric Surgery, The First Affiliated Hospital of Shandong First Medical University & Shandong Provincial Qianfoshan Hospital, Shandong Engineering and Technology Research Center for Pediatric Drug Development, Jinan, Shandong, China

Corresponding authors
Qingyong Liu, lqylbc@163.com
Zuohui Zhao, zhaozuohui@126.com

## ABSTRACT

Lactate dehydrogenase (LDH) is a crucial glycolytic enzyme which mediates the metabolic plasticity of cancer cells, however its clinical significance in renal cell carcinoma (RCC) is poorly understood. Herein, we examined the prognostic significance of the two primary components of LDH, *i.e.*, LDHA and LDHB, in clear cell RCC (ccRCC) patients and further explored their association with immune infiltration in ccRCC. In this study, the expression levels of LDHA and LDHB were examined in ccRCC and adjacent normal tissues by Gene Expression Profiling Interactive Analysis 2 (GEPIA2), UALCAN, and western blotting (WB) analyses, and their prognostic values were estimated in 150 ccRCC and 30 adjacent normal tissues by immunohistochemistry (IHC) analysis. The relationship to immune infiltration of *LDHA* and *LDHB* genes was further investigated using tumor immune estimation resource 2 (TIMER2) and Tumor-Immune System Interactions and DrugBank (TISIDB) databases, respectively. Public databases and WB analyses demonstrated higher LDHA and lower LDHB in ccRCC than in non-tumor tissues. IHC analysis revealed that LDHA and LDHB expression profiles were significantly associated with tumor grade, stage, size, and overall survival (OS). Univariate survival analysis displayed that high grade, advanced stage, large tumor, metastasis, high LDHA, and low LDHB expression were significantly associated with a poorer OS, and multivariate analysis revealed tumor stage and LDHB were identified as independent predictors for OS in patients with ccRCC. Further TIMER2 and TISIDB analyses demonstrated that *LDHA* and *LDHB* expression was significantly related to multiple immune cells and immune inhibitors in over 500 ccRCC patients. These findings revealed that LDHB was an independent favorable predictor, and LDHA and LDHB correlated with tumor immune infiltrates in ccRCC patients,

which indicated LDHA/LDHB could be implicated in the tumorigenesis of ccRCC and might be potential therapeutic targets for patients with ccRCC.

# INTRODUCTION

Renal cell carcinoma (RCC) is the most common type of kidney cancer, with an estimated 79,000 newly diagnosed cases and 13,920 cancer-related deaths in the USA in 2022 (*Siegel et al., 2022*). Clear cell RCC (ccRCC) accounts for approximately 75% of RCC and is refractory to traditional chemotherapy and radiotherapy (*Posadas, Limvorasak & Figlin, 2017*). Radical nephrectomy is the gold standard for localized RCC, which exhibits a good prognosis (about 80% five-year survival rate). However, the prognosis for patients with advanced RCC, especially metastatic RCC (mRCC), remains dismaying, due to their prone to relapse and resistance to conventional therapeutic approaches (*Posadas, Limvorasak & Figlin, 2017*). Although molecular targeted therapy could prolong the survival of advanced RCC patients, further exploration of the molecular mechanisms underlying RCC progression is urgently needed.

Metabolic reprogramming, or altered metabolism, is a critical hallmark of cancers, which facilitates accumulating metabolic intermediates as sources of building blocks (*Jafari et al., 2019*). In addition to aberrant metabolic pathways which cause lipid droplet (LDs) accumulation in ccRCC, recent evidence indicates a link between obesity and ccRCC, and ccRCC has been recognized as a chronic metabolic disease (*Wettersten et al., 2017*). Due to the paradox between uncontrolled cell proliferation and a limited supply of nutrients, cancer cells always reprogram their metabolism, including glucose, protein, nucleic acids, and lipids (*Heravi et al., 2022*). The first recognized and well-known metabolic reprogramming is aerobic glycolysis, which provides bulk intermediated products (including lactate and ATP) for the rapidly proliferating cancer cells even under normoxia (*Wettersten et al., 2017*). Delineating the mechanisms underlying metabolic reprogramming would facilitate understanding RCC's pathophysiology and provide a promising therapy for this heterogeneous tumor.

Lactate dehydrogenase (LDH), a nicotinamide adenine dinucleotide (NAD+) dependent enzyme, is the crucial glycolytic enzyme involved in tumor initiation and metabolism. LDH is a tetrameric enzyme that catalyzes the reversible conversion of pyruvate to lactate, coupled with the oxidation of NADH to NAD+, in the glycolytic pathway (*Urbanska & Orzechowski, 2019*). There are four subtypes of LDH, *i.e.,* LDHA, LDHB, LDHC, and LDHD. Among them, LDHA and LDHB are the significant components of LDH, which mediate the metabolic plasticity of tumor cells. LDHA is abundant in skeletal muscle, which converts pyruvate to lactate and produces NAD+. In contrast, LDHB is predominantly expressed in the brain and heart, which converts lactate to pyruvate for further oxidization (*Ding, Karp & Emadi, 2017*; *Urbanska & Orzechowski, 2019*). LDHC is mainly limited to the testis, while LDHD is universally found in various tissues (*Urbanska & Orzechowski, 2019*).

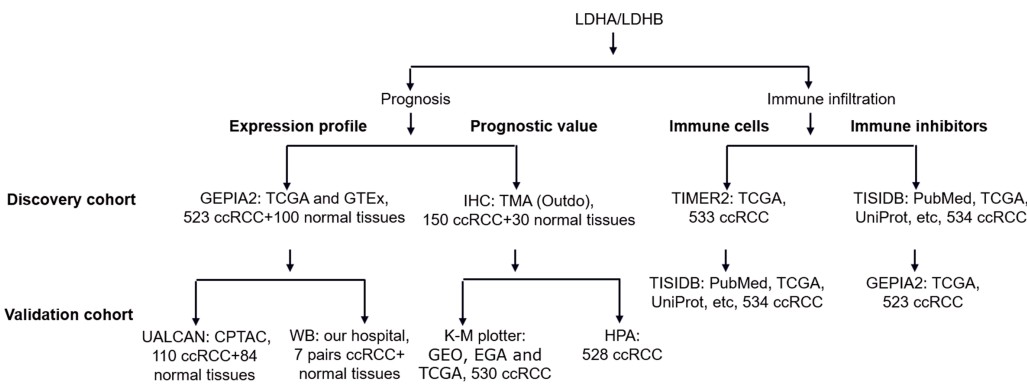

**Figure 1** **The workflow of prognosis and tumor-immune infiltration of LDHA/LDHB in ccRCC.**
Abbreviation: Clinical Proteomic Tumor Analysis Consortium, CPTAC; European Genome-phenome Archive, EGA; Gene Expression Omnibus, GEO; Gene Expression Profiling Interactive Analysis 2, GEPIA2; Genotype-Tissue Expression, GTEx; Human Protein Atlas, HPA; Kaplan–Meier plotter, K–M plotter; The Cancer Genome Atlas, TCGA; Tumor Immune Estimation Resource 2, TIMER2; Tumor-Immune System Interactions and DrugBank, TISIDB; Tissue Microarray, TMA.

Using quantitative proteomics analysis, our previous study identified numerous dysregulated proteins, such as hydroxy acyl-CoA dehydrogenase alpha subunit (HADHA), LDHA, and LDHB, which might be implicated in RCC pathogenesis (*Zhao et al., 2015*). Recent literature reported that the isoenzymes of LDH, including LDHA, LDHC, and LDHD, were significantly correlated with the clinical outcomes of RCC (*Girgis et al., 2014*; *Hua et al., 2017*; *Wang et al., 2018*; *Zhao et al., 2017*). At the same time, the role of LDHB in RCC remains elusive. In addition, increasing evidence demonstrated the tight connection between LDH and tumor immune infiltration (*Ding, Karp & Emadi, 2017*), which still needs further exploration. In the current study, we compared the differential expression of LDHA and LDHB between ccRCC and their adjacent kidney tissues using Gene Expression Profiling Interactive Analysis 2 (GEPIA2), UALCAN and western blotting (WB) analyses, detected their expression in 150 ccRCC and 30 normal kidney samples using immunohistochemistry (IHC) analysis with tissue microarray (TMA), assessed their prognostic role in the same 150 ccRCC patients, and explored the relationship between *LDHA/LDHB* gene expression and immune infiltration in ccRCC using Tumor Immune Estimation Resource 2 (TIMER2) and Tumor-Immune System Interactions and DrugBank (TISIDB) databases, which aimed to investigate the clinical significance of LDHA and LDHB in patients with ccRCC (Fig. 1).

## MATERIALS & METHODS

### Tissue samples and data collection

The ethical committees of The First Affiliated Hospital of Shandong First Medical University approved the research (2016-S017), and the participants signed the informed consent. A total of 157 ccRCC patients who underwent nephrectomy were enrolled in this study. All the ccRCC samples were primary lesions verified using hematoxylin and eosin (HE) staining after surgery (Fig. S1A). Cohort #1 was used to compare LDHA and LDHB expression by

**Table 1  Correlation between LDHA and LDHB expression and clinical characteristics of ccRCC ($n = 150$).**

| Parameters | LDHA staining[a] | | $\chi^2$ | P value | LDHB staining[a] | | $\chi^2$ | P value |
|---|---|---|---|---|---|---|---|---|
| | Low (%) | High (%) | | | Low (%) | High (%) | | |
| Sex | | | | | | | | |
| Male ($n = 107$) | 39(36.45) | 68(63.55) | | | 42(39.25) | 65(60.75) | | |
| Female ($n = 43$) | 17(39.53) | 26(60.47) | 0.125 | 0.724[b] | 15(34.88) | 28(65.12) | 0.246 | 0.618[b] |
| Age | | | | | | | | |
| <60 yrs ($n = 73$) | 34(46.58) | 39(53.42) | | | 23(31.51) | 50(68.49) | | |
| ≥60 yrs ($n = 77$) | 22(28.57) | 55(71.43) | 5.192 | 0.023[b] | 34(44.16) | 43(55.84) | 2.545 | 0.111[b] |
| ISUP grade | | | | | | | | |
| G 1-2 ($n = 103$) | 52(50.49) | 51(49.51) | | | 26(25.24) | 77(74.76) | | |
| G 3-4 ($n = 47$) | 4(5.41) | 43(91.49) | 24.305 | <0.001[b] | 31(65.96) | 16(34.04) | 22.708 | <0.001[b] |
| AJCC stage | | | | | | | | |
| T I ($n = 122$) | 54(44.26) | 68(55.74) | | | 36(29.51) | 86(70.49) | | |
| T II-III ($n = 16$) | 1(6.25) | 15(93.75) | | | 10(62.50) | 6(37.50) | | |
| T III ($n = 12$) | 1(8.33) | 11(91.67) | | 0.001[c] | 11(91.67) | 1(8.33) | | <0.001[c] |
| Tumor size | | | | | | | | |
| <7.0 cm ($n = 119$) | 55(46.22) | 64(53.78) | | | 37(31.09) | 82(68.91) | | |
| ≥ 7.0 cm ($n = 31$) | 1(3.23) | 30(96.77) | | <0.001[c] | 20(64.52) | 11(35.48) | 11.661 | 0.001[b] |
| Metastasis | | | | | | | | |
| Negative ($n = 140$) | 55(39.29) | 85(60.71) | | | 48(34.29) | 92(65.71) | | |
| Positive ($n = 10$) | 1(10.00) | 9(90.00) | | 0.059[c] | 9(90.00) | 1(10.00) | | 0.001[c] |
| Survival rate | | | | | | | | |
| Alive ($n = 122$) | 55(45.08) | 67(54.92) | | | 30(24.59) | 92(75.41) | | |
| Dead ($n = 28$) | 1 (3.57) | 27(96.43) | | <0.001[c] | 27(96.43) | 1(3.57) | | <0.001[c] |

**Notes.**

[a] LDHA or LDHB immunoexpression, scored by evaluating the cytoplasmic staining intensity (0∼3) and frequency (0∼4). According to their expression, they were classified into two groups: low group (cancer scores <5 for LDHA, <6 for LDHB) and high group (scores ≥5 for LDHA, ≥6 for LDHB).

[b] Statistical analyses were performed using Pearson chi-square tests.

[c] Statistical analyses were performed using Fisher's exact test.

WB analysis, which consisted of seven cases of ccRCC and their adjacent kidney tissues between March 2017 and January 2019 [including five males and two females, aged from 43 to 73, International Society of Urological Pathology (ISUP) grading with 1 G1 +4 G2 +2 G3, American Joint Committee on Cancer (AJCC) pathological staging with 4 TI +3 TII]. Cohort #2 was used to examine the expression and prognosis of LDHA and LDHB by IHC assay. TMA was provided by Outdo (Shanghai, China), which included 150 primary ccRCC and 30 adjacent normal tissues. The samples were collected between February 2008 and March 2010. The clinicopathological parameters of the 150 ccRCC patients were recorded, including tumor grade, stage, sizes, metastasis status, follow-up period, and the patient's sex and age (Table 1). The median follow-up period was 32.0 months (4 to 90 months).

## LDH expression analysis

GEPIA2 (http://gepia2.cancer-pku.cn/) database was used to analyze the gene expression of the four subtypes of LDH, *i.e., LDHA, LDHB, LDHC,* and *LDHD,* in 523 ccRCC and

100 standard kidney samples from The Cancer Genome Atlas (TCGA) and Genotype-Tissue Expression (GTEx), as previously described (*Huo et al., 2021*; *Tang et al., 2019*). For correlation analysis, GEPIA2 (http://gepia2.cancer-pku.cn/) from TCGA was also used to validate the relationship between LDHB and the four immunoinhibitors, *i.e.,* VTCN1, TGFBR1, ADORA2A and CD160, in the 523 ccRCC tissues. UALCAN (http://ualcan.path.uab.edu/analysis-prot.html) was used to compare the protein expression level of LDHA, LDHB, LDHC, and LDHD in ccRCC (*Chandrashekar et al., 2022*). The clinical proteomic tumor analysis consortium (CPTAC) module was used, and the total proteins of the four subtypes of LDH were compared between 110 ccRCC and 84 normal tissues, respectively.

## Immunoblotting analysis

As previously described, seven pairs of ccRCC and their adjacent tissues were used for WB analysis (*Li et al., 2021*). After incubating with the corresponding primary antibodies: LDHA (1:1,000, rabbit, #3582; CST, Danvers, MA, USA), LDHB (1:5,000, rabbit, ab53292; Abcam, Cambridge, UK), anti-Tubulin (1:2,000, rabbit, 11224-1-AP; Proteintech, Rosemont, IL, USA), horseradish peroxidase (HRP) conjugated secondary antibodies (1:5,000, rabbit, SA00001-2; Proteintech, Rosemont, IL, USA) were used to visualize the desired proteins. The protein bands were developed by enhanced chemiluminescence (ECL, WBKlS0100; Millipore, Burlington, MA, USA), captured by Gel Image System FluorChem M (ProteinSimple, San Jose, CA, USA), and quantified by Image J software.

## IHC analysis

The IHC analysis was performed on the TMA ($n = 150$) as previously described (*Li et al., 2021*; *Wu et al., 2022a*). The slides were stained with primary antibodies: LDHA (1:400, CST) and LDHB (1:200; Abcam), developed with DAB (ZLI-9017; Zhongshan, China). The immunohistochemical staining was analyzed and reviewed on a microscope(Olympus BX53, Tokyo, Japan) by two pathologists unaware of the disease outcome. As LDHA and LDHB were primarily located in the cytoplasm, immunoexpression was scored by evaluating the cytoplasmic staining intensity (0~3) and frequency (0~4) as previously described (*Yuan et al., 2020*). According to their expression, they were classified into two groups: low group (cancer scores <5 for LDHA, <6 for LDHB) and high group (scores ≥5 for LDHA, ≥6 for LDHB).

## The prognosis analysis

Kaplan–Meier (K–M) plotter (http://www.kmplot.com) was used to validate the prognosis (recurrence -free survival, RFS) of *LDHA* in 530 ccRCC patients (*Nagy, Munkacsy & Gyorffy, 2021*). After being loaded into the database, the log-rank *P*- value and hazard ratio (HR) with 95% confidence intervals (CI) were calculated accordingly.

Human Protein Atlas (HPA) database (http://www.proteinatlas.org) was utilized to analyze and confirm the prognosis (OS) of *LDHB* in 528 ccRCC patients as previously reported (*Fan et al., 2020*). In the HPA database, the best expression cut-off was set as the default, and the prognosis indexes, *i.e.,* K–M plot and log-rank *P*-value, were calculated after ≤150 months follow-up.

## Immune infiltration analysis

TIMER2 (http://timer.cistrome.org/) and TISIDB (http://cis.hku.hk/TISIDB) databases were performed to reveal the relationship of *LDHA/LDHB* with immune infiltration in ccRCC, as previously described (*Li et al., 2020*; *Ru et al., 2019*). TIMER2 evaluated the abundance of eight tumor-infiltrating immune cells (TIIC) subsets, *i.e.,* B cells, cancer-associated fibroblast, CD4+ T cells, CD8+ T cells, dendritic cells, endothelial cells, macrophages, and neutrophils, in the ccRCC cohort ($n = 533$). The expression data were log2 transcripts per million (TPM) transformed, Spearman was selected for correlation analysis, and multiple algorithms, including TIMER, OBERSORT, XCELL, and EPIC, were applied for immune infiltration estimations. TISIDB elucidated the correlations between *LDHA/LDHB* expression and abundance of tumor-infiltrating lymphocytes (TILs) & immune inhibitors in ccRCC ($n = 534$).

## Statistical analysis

SPSS 21.0 software (SPSS Inc., Chicago, IL, USA) was used for statistical analysis. Paired student's $t$-test was performed for WB analysis to evaluate LDHA and LDHB expression. For IHC analysis, the Pearson chi-square test or Fisher's exact test was used to assess the associations between LDHA/LDHB expression and clinicopathological parameters, the K–M survival curve was utilized to calculate overall death, and Cox proportional hazard regression analysis was used to analyze the risk factors for ccRCC patients. For TIMER and TISIDB analyses, Spearman's correlations analysis was performed to estimate the correlation between *LDHA/LDHB* and tumor immune infiltrates. $P < 0.05$ was considered statistically significant.

# RESULTS

## LDHA and LDHB expression in ccRCC

First, we detected LDH expression levels between ccRCC and adjacent normal kidney tissues. GEPIA2 database was performed to compare the transcriptional profile of the four subtypes of LDH, *i.e., LDHA, LDHB, LDHC, and LDHD,* in 523 ccRCC and 100 normal kidney tissues. The result demonstrated that *LDHA* mRNA was higher in cancerous than normal tissues ($P < 0.05$), *LDHB* and *LDHD* mRNA was lower ($P < 0.05$, and $P < 0.05$, respectively), and *LDHC* was unchanged in cancerous compared with normal tissues ($P > 0.05$, Fig. 2A, Fig. S1b). In addition, we assessed their protein expression using the CPTAC dataset. Consistently, it demonstrated higher LDHA, lower LDHB and LDHD, and stable LDHC expression in 110 ccRCC than 84 normal kidney tissues ($P < 0.001$, $P < 0.001$, $P < 0.001$, $P > 0.05$, Fig. 2B, Fig. S1c). LDHC is the testis-specific isoform, LDHD is universally expressed in various tissues, and their prognostic values in ccRCC have been reported previously (*Hua et al., 2017*; *Wang et al., 2018*). Herein, we focused on the significant components of LDH, *i.e.,* LDHA and LDHB, in ccRCC. WB analysis showed that LDHA was significantly up-regulated and LDHB was down-regulated in the seven pairs of ccRCC than their non-cancerous specimens ($P < 0.001$, $P < 0.001$, respectively, Fig. 2C), which was consistent with GEPIA2 and ULACAN analyses.

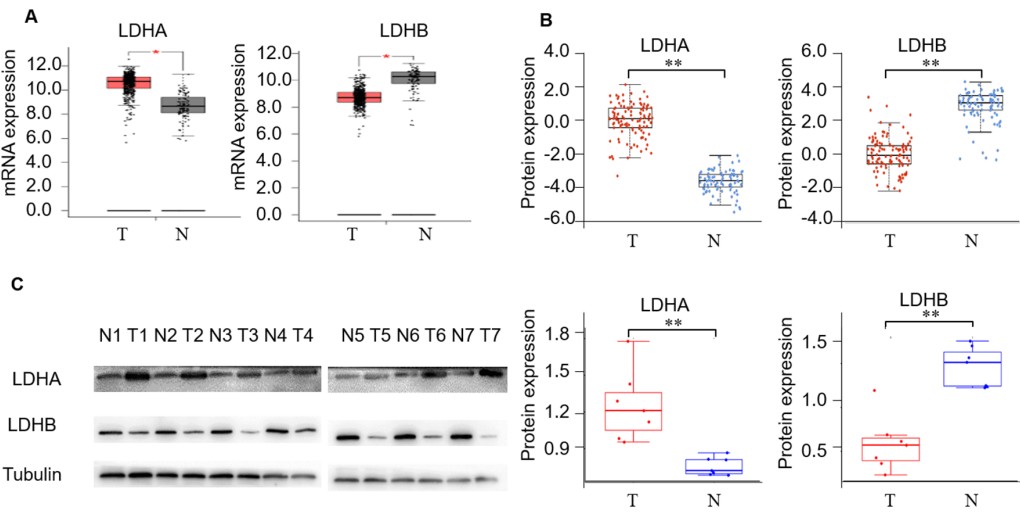

**Figure 2  The expression profiling of LDHA and LDHB in ccRCC tissues.** (A) The mRNA expression levels of LDHA and LDHB in 523 ccRCC and 100 adjacent normal kidney tissues (GEPIA2, ANOVA test). (B) The protein expression levels of LDHA and LDHB in 100 ccRCC and 84 adjacent normal kidney tissues (UALCAN, ANOVA test). (C) LDHA and LDHB protein expression in seven pairs ccRCC and their adjacent kidney tissues (WB, paired $t$-test). T, ccRCC; N, normal kidney tissues. $*P < 0.05$, $**P < 0.01$.

## Enhanced LDHA and decreased LDHB are associated with tumor aggressiveness of ccRCC

Subsequently, we evaluated LDHA/LDHB expression and its correlation with clinicopathological features in ccRCC patients (Table 1). As ccRCC is characterized by amounts of LDs, *i.e.*, lipid and glycogen, in the cytoplasm. We first checked the tissue morphology using HE staining, which demonstrated normal kidney tubule epithelial cells and glomerular endothelial cells in adjacent non-neoplastic tissues, and compact nests of tumor cells with clear cytoplasm (enriched LDs) separated by delicate vasculature (Fig. S1a). IHC analysis showed that solid cytoplasm staining for LDHA expression was seen in the malignant cells of the kidney. In contrast, relatively weak staining for LDHB expression was seen in the neoplastic cells, compared with the adjacent kidney epithelial cells (Fig. 3). Specifically, more important positive signaling with LDHA was monitored in 94 (62.67%) cases of ccRCC tissues, and weaker staining was examined in 56 (37.33%) cases, respectively. The expression level of LDHA was higher in large tumors (≥ 7.0 cm) than in small ones (<7.0 cm); the difference was statistically significant ($P < 0.001$, Fisher's exact test). Simultaneously, enhanced LDHA expression (≥5 scores) was positively associated with high grade (grade 3–4, $P < 0.001$, Pearson chi-square test), advanced stage (stage II–III, $P = 0.001$, Fisher's exact test, Fig. S2a), older age (≥60 years, $P = 0.023$, Pearson chi-square test) and low overall survival (OS) rate ($P < 0.001$, Fisher's exact test), which was consistent with a previous study (*Girgis et al., 2014*). There was no significant association between LDHA expression and patients' sex or metastatic status ($P = 0.724$, Pearson chi-square test, $P = 0.059$, Fisher's exact test, respectively). As for LDHB, its reduced expression (<6 scores) was significantly associated with tumor grade ($P < 0.001$, Pearson chi-square test),

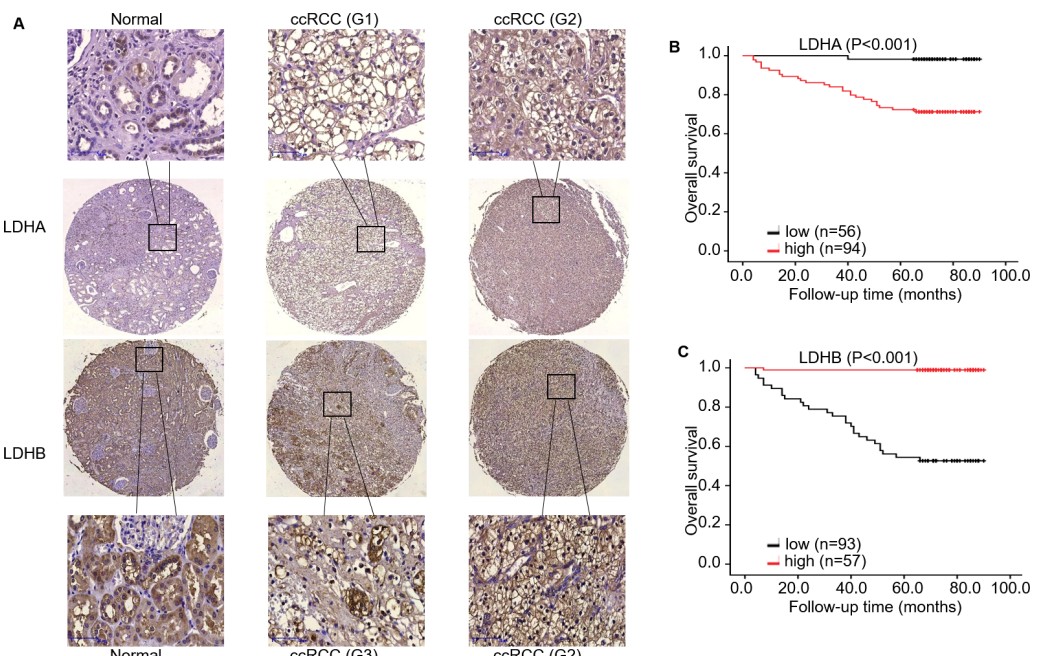

**Figure 3** **The expression and prognosis of both LDHA and LDHB in 150 ccRCC tissues.** (A) Representative immunostaining photomicrographs of LDHA and LDHB expression in ccRCC tissues (IHC). Staining signals displayed cytoplasmic localization of LDHA and LDHB in adjacent normal kidney (left) and ccRCC tissues (middle: low expression, right: high expression). G: ISUP grade, original magnification 200 ×; bars, 50 μm. Kaplan–Meier survival curves demonstrated overall survival of 150 patients with ccRCC, according to LDHA (B) and LDHB (C) staining.

stage ($P < 0.001$, Fisher's exact test, Fig. S2A), size ($P = 0.001$, Pearson chi-square test), metastasis ($P = 0.001$, Fisher's exact test), and survival rate ($P < 0.001$, Fisher's exact test), instead of patients' sex or age ($P = 0.618$, $P = 0.111$, respectively, Pearson chi-square test). The further bioinformatic analysis demonstrated that LDHA showed a trend of positively associated with RFS ($P = 0.100$, Fig. S2b, K–M plotter) and *LDHB* expression was inverse associated with OS ($P = 0.004$, Figure S2c, HPA), which validated our prognostic analysis using IHC. Collectively, these data revealed that high LDHA / low LDHB expression was positively associated with malignant behaviors such as pathological stage and tumor size, and negatively associated with OS, which indicated that high LDHA / low LDHB could be an indicator of tumor aggressiveness for patients with ccRCC. Furthermore, this is the first time to evaluate LDHB prognosis in ccRCC.

## LDHB, but not LDHA, is an independent predictor of OS in patients with ccRCC

To investigate the impact of LDHA/ LDHB expression on tumor prognosis, survival analysis was utilized to evaluate the correlation of their expression with the survival of ccRCC patients ($n = 150$). During the follow-up period, K–M survival analysis manifested that the OS rate with high LDHA expression was significantly lower than that with low

**Table 2** Univariate and multivariate survival analysis of overall survival ($n = 150$).

| Parameters | Univariate[a] HR (95% CI)[b] | P- value | Multivariate[a] HR (95% CI)[b] | P- value |
|---|---|---|---|---|
| Sex | 0.386 (0.134–1.111) | 0.078 | | |
| Age | 1.823 (0.841–3.949) | 0.128 | | |
| Grade (G3-4)[c] | 4.911 (2.264–10.650) | <0.001 | | |
| Stage (TII-III)[d] | 8.346 (3.931–17.722) | <0.001 | 3.918 (1.827–8.400) | <0.001 |
| Size (≥7.0 cm) | 5.004 (2.380–10.520) | <0.001 | | |
| Metastasis | 9.046 (3.803–21.515) | <0.001 | | |
| High LDHA | 18.653 (2.534–137.309) | 0.004 | | |
| High LDHB | 0.017 (0.002–0.128) | <0.001 | 0.025 (0.003–0.186) | <0.001 |

Notes.

[a]Statistical analysis by Cox proportional hazards regression model.

[b]Abbreviation: HR, hazard ratio; CI, confidence interval.

[c]For grade: 1, 2 vs 3–4.

[d]For stage: I vs II-III.

expression (log-rank $=16.154$, $P < 0.001$), while low LDHB expression was markedly correlated with high OS rate(log-rank $= 53.048$, $P < 0.001$, Fig. 3).

Then univariate Cox regression analysis manifested that high LDHA expression was associated with poor prognosis for OS in ccRCC patients (HR 18.653, 95% CI $= 2.534–137.309$, $P = 0.004$, Table 2). Simultaneously, it demonstrated that large tumors (HR 5.004, 95% CI $= 2.380–10.520$, $P < 0.001$), high histological grade (HR 4.911, 95% CI $= 2.264–10.650$, $P < 0.001$), advanced pathological stage (HR 8.346, 95% CI $= 3.931–17.722$, $P < 0.001$), metastasis (HR 9.046, CI $= 3.803–21.515$, $P < 0.001$) and low LDHB expression (HR 0.017, 95% CI $= 0.002−0.128$, $P < 0.001$), were all correlated with a shorter OS rate. Moreover, no association existed between OS and patients' sex or age ($P = 0.078$, $P = 0.128$, respectively). Furthermore, multivariate Cox regression analysis identified that pathological stage (HR 3.918, 95% CI $= 1.827−8.400$, $P < 0.001$) and LDHB (HR 0.025, 95% CI $= 0.003−0.186$, $P < 0.001$) were recognized as independent prognostic indicators for OS in ccRCC patients. In contrast, sex, age, grade, tumor size, metastasis, and LDHA expression were not identified as independent predictors.

## TIMER2 and TISIDB analyses reveal the close relationship between LDHA/LDHB and immune infiltrates in ccRCC

We further performed data mining based on the expression and prognosis analysis of LDHA and LDHB in ccRCC. We investigated the correlation between the two subtypes of LDH, especially LDHB, and immune features, such as immune cells and immunomodulators, in ccRCC using TIMER2 and TISIDB databases. TIMER2 analysis displayed that *LDHB* gene expression was significantly associated with infiltration of seven TIIC subsets, *i.e.,* B cell (rho $= 0.109$, $P = 0.019$), cancer-associated fibroblast (rho $= −0.116$, $P = 0.013$), CD4+ T cell (rho $= 0.411$, $P < 0.001$), CD8+ T cell (rho $= −0.249$, $P < 0.001$), endothelial cell (rho $= −0.234$, $P < 0.001$), macrophage (rho $= 0.247$, $P = 0.001$), and neutrophil (rho $= 0.167$, $P < 0.001$), except dendritic cell (rho $= 0.021$, $P = 0.651$) in the 533 ccRCC samples (Fig. 4). There was also a tight connection between LDHA and infiltration of TIICs, including B cell

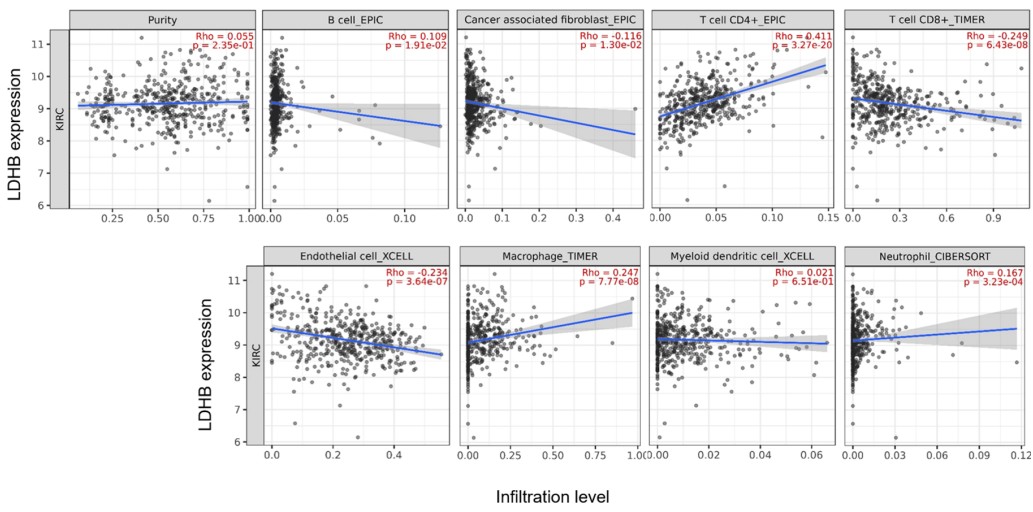

**Figure 4** **Correlation between LDHB expression and tumor-infiltrating immune cells in 533 ccRCC patients (TIMER2).** The infiltration levels of the eight TIIC subsets, *i.e.,* B cell (EPIC), cancer associated fibroblast (EPIC), CD4+ T cell (EPIC), CD8+ T cell (TIMER), endothelial cell (XCELL), macrophage (TIMER), myeloid dendritic cell (XCELL) and neutrophil (OBERSORT).

(rho = 0.129, $P = 0.006$), cancer-associated fibroblast (rho = $-0.150$, $P = 0.001$), CD4+ T cell (rho = $-0.201$, $P < 0.001$), CD8+ T cell (rho = $-0.243$, $P < 0.001$), endothelial cell (rho = 0.136, $P = 0.003$), dendritic cell (rho = 0.162, $P < 0.001$), and neutrophil (rho = 0.164, $P < 0.001$), except macrophage (rho = 0.086, $P = 0.066$) in the same ccRCC samples (Fig. S3). Simultaneously, TISIDB analysis revealed that LDHB expression was associated with the abundance of numerous TILs in the 534 ccRCC cases (Fig. 5). To be specific, LDHB expression was positively correlated with the abundance of immature dendritic cells(iDC, rho =0.363, $P < 0.001$) and activated dendritic cell (Act_DC, rho =0.192, $P < 0.001$), and inversely associated with the abundance of effector memory CD8+ T cell (Tem_CD8, rho =$-0.366$, $P < 0.001$) and natural killer T cell (NKT, rho =$-0.228$, $P < 0.001$). Similarly, LDHA expression was positively related to the abundance of immature dendritic cells (iDC, rho =0.383, $P < 0.001$) and central memory CD8$^+$T cell (Tcm_CD8, a subtype of CD8$^+$ T cell, rho =0.301, $P < 0.001$), and negatively correlated with the abundance of eosinophil cell (rho =$-0.273$, $P < 0.001$) and activated B cell (Act_B, rho =$-0.124$, $P = 0.004$, Fig. S4).

Moreover, we investigated the relationship between LDHB expression and the abundance of 24 immune inhibitors in ccRCC (Fig. 6). Specifically, the greatest positively correlated immunoinhibitors included B7 homolog 4 (B7-H4, or VTCN1, rho =0.235, $P < 0.001$), transforming growth factor- $\beta$ receptor type I (TGFBR1, rho =0.112, $P = 0.009$), and the negatively associated immunoinhibitors were adenosine A2a receptor (A2AR, ADORA2A, rho =$-0.387$, $P < 0.001$) and CD160 (rho =0.339, $P < 0.001$) in ccRCC. As for LDHA, the four immunoinhibitors with the greatest correlations included interleukin-10 receptor B (IL10RB, rho =0.412, $P < 0.001$), indoleamine 2,3-dioxygenase 1 (IDO1, rho =0.154, $P < 0.001$), CD112(PVRL2, rho =0.090, $P = 0.039$) and CD160 (rho =0.083, $P = 0.055$)

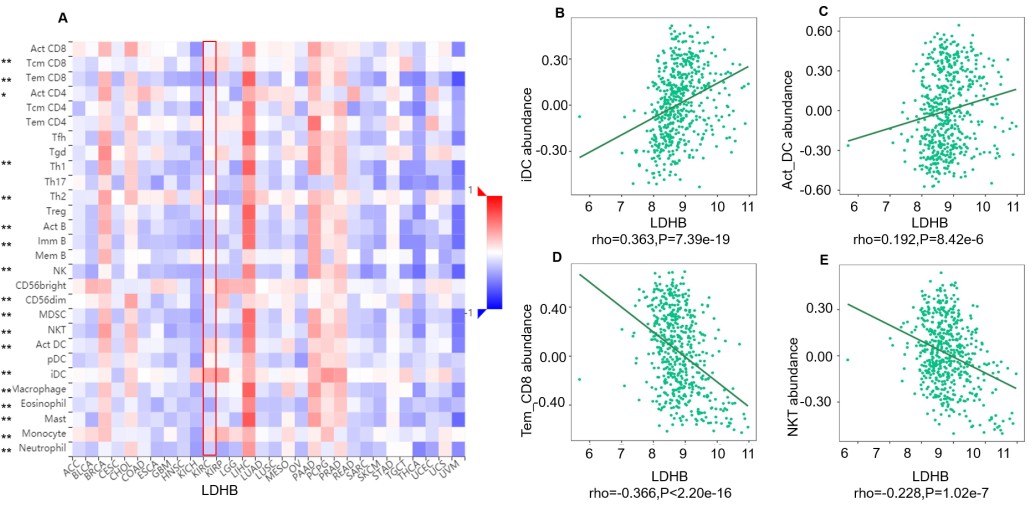

**Figure 5  Correlation between LDHB expression and lymphocytes in 534 ccRCC patients (TISIDB).**
(A) The pan-cancer analysis of relationship between LDHB expression and abundance of the 28 tumor-infiltrating lymphocytes (TILs). The top four lymphocytes either positive (B: iDC cell, C: Act_DC cell) or negative (D: Tem_CD8 cell, E: NKT) correlation with LDHB expression in ccRCC patients. Tem_CD8: effector memory CD8+ T cell, Tcm_CD8: central memory CD8+ T cell. *$P < 0.05$, **$P < 0.01$.

in ccRCC (Fig. S5). Furthermore, GEPIA2 analysis validated the tight relationship between *LDHB* and the four immunoinhibitors in 523 ccRCC tissues (Fig. S6). The above results implied that both LDHA and LDHB might be involved in regulating the immune infiltrates in ccRCC patients, which was consistent with previous reports (*Ding, Karp & Emadi, 2017*).

## DISCUSSION

Aerobic glycolysis, or the Warburg effect, is the well-known and continually validated metabolic reprogramming of cancer. LDH is a critical enzyme involved in glycolysis and carcinogenesis, while its clinical significance in RCC has yet to be fully elucidated. Our previous study found numerous differentially expressed metabolic enzymes, such as HADHA, LDHA, and LDHB, in ccRCC tissues, implying the dysregulated metabolic pathways in the pathogenesis of ccRCC (*Zhao et al., 2015*). In the present study, we recapitulated that the major components of LDH, *i.e.,* LDHA and LDHB, were promising indicators for prognosis and immune infiltration in ccRCC (Fig. 1). Our study validated the aberrant expression of LDHA and LDHB in ccRCC tissues, *i.e.,* LDHA was up-regulated, and LDHB was down-regulated in ccRCC, consistent with previous reports (*Girgis et al., 2014*). Then, retrospective IHC analysis revealed that the expression levels of LDHA and LDHB were significantly associated with tumor grade, stage, size, and OS, which indicated that enhanced LDHA and decreased LDHB were positively correlated with ccRCC aggressiveness. Subsequently, survival analysis revealed that LDHB, instead of LDHA, was recognized as an independent prognostic indicator for OS in 150 ccRCC patients. Further TIMER2 and TISIDB databases analysis manifested the close relationship between

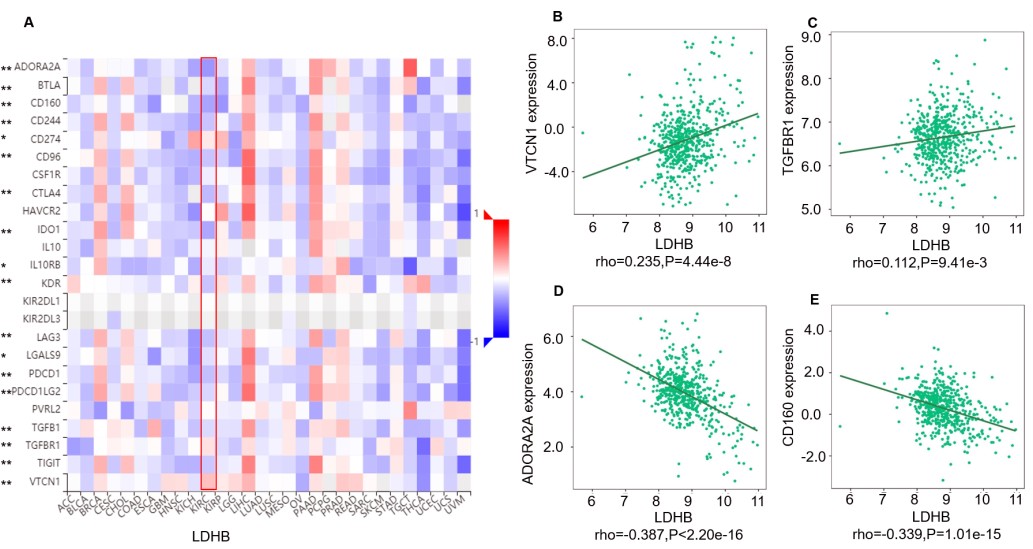

**Figure 6** **Correlation between LDHB expression and immunoinhibitors in 534 ccRCC patients (TISIDB).** (A) The pan-cancer analysis of relationship between LDHB expression and abundance of the 24 immunoinhibitors. The top four immunoinhibitors (VTCN1 (B), TGFBR1 (C), ADORA2A (D) and CD160 (E)) either positively or negatively correlated with LDHB expression in ccRCC patients. *$P < 0.05$, **$P < 0.01$.

LDHA/LDHB expression and immune infiltrates (including immune cells and immune inhibitors) in >500 ccRCC patients, which indicates the complex tumor microenvironment (TME) of ccRCC. To our knowledge, this is the first time to elucidate the clinical significance of LDHB in ccRCC patients, which revealed that LDHB could be a favorable prognostic factor and might regulate multiple immune features in ccRCC. Further studies are needed to explore the detailed mechanism underlying LDHB in ccRCC carcinogenesis.

Metabolic reprogramming, or metabolic plasticity, is an essential hallmark of cancers. It enables rapidly proliferating cancer cells to meet their needs for augmented energetics and building components. Emerging evidence illuminates the perturbed metabolic pathways which could control tumor energetics and biosynthesis in cancer, especially in ccRCC (*Wettersten et al., 2017*). Such aberrant metabolic pathways in ccRCC could provide opportunities to discover novel diagnostic biomarkers and therapeutic targets, which might improve the overall prognosis of ccRCC patients (*Wettersten et al., 2017*). In non-neoplastic or normal cells, glucose is converted to pyruvate, which undertakes oxidative phosphorylation (OXPHOS) for energy production under normoxia. Cancer cells predominantly produce energy and lactate by aerobic glycolysis, regardless of oxygen availability. ccRCC, characterized by high glucose uptake and enhanced levels/activities of glycolytic enzymes, such as hexokinase and LDHA, has been aptly labeled as a metabolic disease (*Wettersten et al., 2017*).

LDH isoenzymes are NAD+ dependent metabolic enzymes that are reportedly linked to RCC pathogenesis (*Girgis et al., 2014*; *Hua et al., 2017*; *Wang et al., 2018*; *Zhao et al., 2017*). LDH is the critical enzyme involved in aerobic glycolysis, which mediates metabolic
plasticity through the bidirectional conversion of pyruvate and lactate. LDHA converts pyruvate to lactate and NADH to NAD+ in anaerobic conditions, whereas LDHB possesses a higher affinity for lactate, preferentially converting lactate to pyruvate when oxygen is abundant. As LDHA and LDHB participate in tumor cell metabolism and adaptation to detrimental cellular conditions, these enzymes are reportedly involved in tumor pathogenesis and progression (*Urbanska & Orzechowski, 2019*). Except for LDHA and LDHB, LDHC and LDHD are expressed in various cancers (*Urbanska & Orzechowski, 2019*). Previous studies showed elevated serum LDH was an unfavorable prognostic factor in RCC, especially metastatic RCC (*Zhang et al., 2020*). LDHA is overexpressed in various neoplastic tissues, and enhanced LDHA expression is associated with worse prognosis of patients with brain, liver, lung, and kidney tumors (*Urbanska & Orzechowski, 2019*). Through IHC analysis, Girgis reported that overexpressed LDHA was associated with poor prognosis (including disease-free survival and OS) in 385 ccRCC patients, which validated its OS in an independent 170 ccRCC patients from TCGA databases (*Girgis et al., 2014*). This was a large-scale specimen, but it only evaluated the prognosis of LDHA. Zhao observed that elevated LDHA predicted worse survival in 43 ccRCC patients using IHC staining. LDHA knockdown attenuated tumor metastasis by inhibiting epithelial-mesenchymal transition (EMT) (*Zhao et al., 2017*). Similarly, *Wang et al. (2017)* demonstrated the oncogenic role of LDHA in RCC cells, which indicated that LDHA might be a potential therapeutic target in RCC. As for LDHB, Wang observed that LDHB expression was higher in pancreatic cancer tissues using IHC analysis, and its expression was negatively correlated with prognosis (OS) in 50 pancreatic cancer patients (*Wang et al., 2022*). Interestingly, Wu found that LDHB expression was lower in glioma, and LDHB was identified as a protective factor using Chinese Glioma Genome Altas (CGGA) and TCGA databases (*Wu et al., 2022b*). The expression and prognosis of LDHB in cancer are controversial, and the clinical value of LDHB in ccRCC is unclear. Cancer–testis antigens(CTAs) are expressed in the testis and various cancers, and they are considered promising targets for early diagnosis and immunotherapy for cancers. As a member of CTAs, LDHC level was significantly up-regulated in RCC tissues, and the patients with positive LDHC expression had a shorter progression-free survival (PFS) in 133 RCC. Further *in vitro* experiments displayed that LDHC could promote RCC progression through EMT, indicating the oncogenic role of LDHC in RCC (*Hua et al., 2017*). Wang found that *LDHD* genes expression was considered to be a favorable predictive of the prognosis (OS) of ccRCC patients from TCGA ($n = 509$) and Fudan University Shanghai Cancer Centre (FUSCC, $n = 192$) cohorts, which indicated *LDHD* might be involved in ccRCC pathogenesis (*Wang et al., 2018*). Herein we identified LDHB as a favorable prognostic marker that closely correlated with immune infiltrates in ccRCC, and this is the first time to elucidate the clinical significance of LDHB in ccRCC to the best of our knowledge. LDHB converts lactate to pyruvate and produces NADPH, thus providing sufficient energy for tumor cell proliferation while avoiding the accumulation of lactate, which indicates it could be the potential therapeutic target for ccRCC, especially metastatic ccRCC.

LDHB downregulation has been observed in various types of cancer, and it is important to understand the underlying mechanisms behind this phenomenon. Epigenetic

modifications, such as DNA methylation and histone deacetylation, have been shown to play a role in regulating LDHB expression (*De Mello et al., 2017*). DNA methylation is an epigenetic modification that involves the addition of a methyl group to the cytosine residue of CpG dinucleotides. Hypermethylation of the LDHB promoter region has been reported in several types of cancer, including gastric cancer, hepatocellular carcinoma, and pancreatic cancer (*Maekawa et al., 2003*). In these cases, hypermethylation of the promoter region results in the silencing of LDHB expression. Histone deacetylation is another epigenetic modification that can lead to gene silencing. The histone deacetylase inhibitor trichostatin A has been shown to upregulate LDHB expression in breast cancer cells (*Rodrigues et al., 2015*). Dysregulated microRNA expression has also been implicated in the downregulation of LDHB expression (*Frank et al., 2021*). MicroRNAs are small RNA molecules that negatively regulate gene expression by binding to the 3′ untranslated region (UTR) of target mRNAs and causing their degradation or translational repression (*Ali Syeda et al., 2020*). Several studies have identified specific microRNAs that target LDHB. For example, miR-375 has been shown to downregulate LDHB expression in breast cancer cells (*Frank et al., 2021*). In summary, the downregulation of LDHB in cancer might be due to epigenetic modifications such as DNA methylation and histone deacetylation, as well as dysregulated microRNA expression. Understanding the mechanisms behind LDHB downregulation may help to identify potential therapeutic targets for cancer.

Recent literature elaborates that the intermediates of cancer metabolism could be essential in regulating the proliferation, differentiation, and function of immune cells, which gives birth to immunometabolism (*Shyer, Flavell & Bailis, 2020*). Cancer cells, immune cells, secreted factors, and extracellular matrix proteins collectively constitute the complex dynamics of TME. Cancer cells can suppress the anti-tumor immune response by competing for and depleting essential nutrients and reducing the metabolic fitness of TIICs. Like cancer cells, TIICs require nutrients derived from the TME to support their proliferation and differentiation. They also undergo metabolic reprogramming. During aerobic glycolysis, hypoxia, low pH, high levels of reactive oxygen species (ROS), and lactate accumulation are prevalent in the TME, which have a deleterious effect on the immune function (*Harmon, O'Farrelly & Robinson, 2020*). Thus, the higher lactate content and the accompanying acidified TME will suppress immune cell function and abrogate immunosurveillance of cancer, ultimately leading to immune escape and cancer progression (*Xia et al., 2021*). In particular, lactate accumulation could deplete Teff cells and affect Treg cell infiltration, thus promoting the formation of an inhibitory immune microenvironment (*Wang et al., 2021*). Interestingly, *Singer et al. (2011)* found that the increased GLUT-1 expression was correlated with a decrease in the numbers of infiltrating CD3+ and CD8+ T cells in 80 cases of ccRCC, suggesting that GLUT-1 might suppress the immune system in ccRCC. In the current study, we found there was a close correlation between LDHA and LDHB expression levels and multiple TIIC subsets, *i.e.,* B cells, cancer-associated fibroblast, CD4+ T cells, CD8+ T cells, endothelial cells, and neutrophils, and massive immunoinhibitors such as VTCN1 (Figs. 4 and 6). We identified LDHB as a favorable prognostic marker, and LDHA/LDHB was correlated with immune infiltrates in ccRCC,

which confirmed the tight connection between immune and metabolism. Furthermore, the underlying molecular mechanism of immunometabolism still needs further investigation.

There are some limitations of this study. The first one is the limited sample size of the IHC analysis. Most of them are localized lesions, which need more specimens and prolonged follow-up periods to validate our results. The second shortcoming is that only one primary outcome, *i.e.,* OS, is analyzed in the enrollment; CSS and RFS are also needed to clarify the clinical role of LDHA/LDHB in ccRCC. The third limitation is the lack of independent validation at protein level either through IHC or flow cytometry of LDHA/LDHB levels in tissue with immune cell subtypes. Finally, it should be marked that the detailed mechanism between LDHA/LDHB and tumor immune needs further clarification.

## CONCLUSION

In the current study, we detected LDHA and LDHB expression using public databases and WB analyses, explored their prognostic role in ccRCC using TMA, then revealed the tumor-immune interaction of LDHA/LDHB in ccRCC using TIMER2 and TISIDB databases. These findings revealed that LDHB was an independent predictor of favorable survival. Both LDHA and LDHB were associated with tumor immune infiltrates in ccRCC patients, which suggested LDHA/LDHB could be implicated in the tumorigenesis of ccRCC and might be potential therapeutic targets for patients with ccRCC.

### Funding

This work was supported by the Cultivation Foundation of The First Affiliated Hospital of Shandong First Medical University (QYPY2019NFSC0601), the Shandong Medical and Health Science and Technology Development Project (2016WS0481), the Clinical Medicine Innovation Program of Jinan City (202019125), the Shandong Provincial Nature Science Foundation (ZR2020QH240), and the National Nature Science Foundation of China (NSFC82002719). The funders had no role in study design, data collection and analysis, decision to publish, or preparation of the manuscript.

### Grant Disclosures

The following grant information was disclosed by the authors:
Cultivation Foundation of The First Affiliated Hospital of Shandong First Medical University: QYPY2019NFSC0601.
Shandong Medical and Health Science and Technology Development Project: 2016WS0481.
Clinical Medicine Innovation Program of Jinan City: 202019125.
Shandong Provincial Nature Science Foundation: ZR2020QH240.
National Nature Science Foundation of China: NSFC82002719.

### Competing Interests

The authors declare there are no competing interests.

## Author Contributions

- Jie Chen performed the experiments, analyzed the data, prepared figures and/or tables, and approved the final draft.
- Fei Wu analyzed the data, authored or reviewed drafts of the article, and approved the final draft.
- Yehua Cao analyzed the data, authored or reviewed drafts of the article, and approved the final draft.
- Yuanxin Xing performed the experiments, prepared figures and/or tables, and approved the final draft.
- Qingyong Liu conceived and designed the experiments, prepared figures and/or tables, and approved the final draft.
- Zuohui Zhao conceived and designed the experiments, prepared figures and/or tables, authored or reviewed drafts of the article, and approved the final draft.

## Human Ethics

The following information was supplied relating to ethical approvals (i.e., approving body and any reference numbers):

Ethical Committee of The First Affiliated Hospital of Shandong First Medical University (2016-S017)

## Data Availability

The raw measurements are available in the Supplementary Files.

## Supplemental Information

Supplemental information for this article can be found online at http://dx.doi.org/10.7717/peerj.15749#supplemental-information.

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
