# Peer review of "The novel role of LDHA/LDHB in the prognostic value and tumor-immune infiltration in clear cell renal cell carcinoma"

_PeerJ, doi:10.7717/peerj.15749_

## Round 0.1 · original submission · Minor Revisions

As you can see, all four reviewers provided very supportive comments. However, they also indicated that some amendments are needed. Therefore, please address the concerns of all reviewers and revise your manuscript accordingly.

Reviewer 1 ·

Basic reporting

The language is clear and well written. The authors have provided enough references to support their statements made in the introduction as well as discussions. There is right amount of background information available. All necessary figures/tables and raw data are shared clearly.

Experimental design

The research is meaningful and needed, it has been well studied in other types of cancer, however there has been no clear correlation btw LDHB and tumor immune response. But this research would serve as a good starting point to further explore this relationship. The methods are clearly written and have detailed information for future studies.

Validity of the findings

The rationale is clearly stated and has been validated by providing substantial data and most of the results have been statistically valid. The authors have used normal tissues as controls. The conclusions are well stated and have also mentioned the drawback/limitations of the manuscript

Annotated reviews are not available for download in order to protect the identity of reviewers who chose to remain anonymous.

Reviewer 2 ·

Basic reporting

The research manuscript “The novel role of LDHA/LDHB in the prognostic value and tumor-immune infiltration in clear cell renal cell carcinoma” by Chen et al investigates prognostic significance of LDHA/LDHB, the critical components of glycolytic enzyme LDH in clear cell renal cell carcinoma. Authors have investigated expression of LDHA and LDHB in cancerous and normal tissues by various experimental techniques including, expression profiling by (GEPIA2), western blotting and immunohistochemistry. Research findings from the study suggests a significantly elevated mRNA expression of LDHA in cancerous samples compared to the normal tissues. Expression of LDHB and LDHD mRNA was found to be low, whereas LDHC expression level didn’t had any change in cancerous samples compared with normal tissues. It’s clinically interesting to observe direct correlation between tumor size, grades and LDHA expression.

1. Line 28-29 is complex and needs fragmentation for improved understanding and clarity in the flow. Manuscript at several places need improvement in English. Kindly, recheck the manuscript for grammar and flow.
2. The study hasn’t explored the mechanism behind the down-regulated expression of LDHB. It would add value to explain the possible cause in the discussion section.

Experimental design

The study design is meaningful and satisfactory for the most parts. The research findings are well supported by figures and tables. The lack of data regarding the functional mechanism involved needs explanation in the text.

Validity of the findings

The study is of high clinical relevance as it also shows LDHB as an independent predictor of overall survival in patients with ccRCC. However, as already mentioned by the authors the study lacks the mechanism behind the deregulated expression and future studies are warranted to explore the mechanism further. The study can be accepted for the publication based on clinical significance. However, a few minor points need to be explained.

Additional comments

Can be accepted after a minor revision.

·

Basic reporting

no comment

Experimental design

Methods related suggestions - Please update the 'Statistical Analysis' section addressing the major comments 4 and 5, included below

Validity of the findings

Please address all comments included in the additional comments section (major comment 8 ) below

Additional comments

In this manuscript, authors have investigated the expression profiles of metabolic enzymes LDHA and LDHB in clear cell renal cell carcinoma (ccRCC) compared to healthy renal tissue. Authors found, using IHC, WB as well as public databases (TCGA) that protein expression of LDHA was positively associated with poor survival and that in case of LDHB, protein expression was negatively associated with overall survival in ccRCC tissue. Additionally, authors also observed that LDHB was negatively associated with tumor aggressiveness and could predict survival in ccRCC patients independently of other factors. This is the most important finding of the paper.
Using additional data mining, authors report a strong correlation between LDHA and CD4+ T cells as well as LDHB and CD4+ T cells in addition to other immune cell types.
Overall, this is a fairly well written manuscript which is within the scope of the journal and fairly scientifically sound as it employs multiple experimental methods (both WB and IHC in addition to public datasets) to confirm its major finding. I have the following comments which can help improve the manuscript further -

Major comments

1. Prior to conducting WB or IHC, what methods (or cancer markers) were used to confirm the tissues imaged were indeed cancerous? Similarly, what markers were used to confirm that adjacent kidney cells were non-cancerous

2. Though the association with metastatic status is not statistically-significant, it may yet be biologically relevant and the authors should note this either in results or discussion.

3. How do authors reconcile the conflicting results of the correlation of CD8+ T cells and LDHA expression between TISIDB and TIMER analysis?

4. In some cases (for e.g., the WB samples), patients have paired data; did authors use appropriate paired tests to assess statistical differences? It is not clear.

5. In Results, lines 241 onwards, would it be prudent to use a multiple testing correction here?
Clearly, CD4+T cells correlate highly with LDHB, more than any other immune cell type and a multiple comparison p-value correction would capture this. I recommend authors to verify this analysis by a statistician.

6. Another limitation is the lack of independent validation at protein level either through IHC or IF or flow cytometry of LDHA/LDHB levels in tissue with immune cell subtypes. Authors should address this in the Discussion.

7. Why did the authors use DAB and not ECL for WB. ECL is much more sensitive than DAB which is also very toxic.
Please also mention In Methods how the developed Western Blots were imaged?
Did authors explore other methods of testing tumor tissue for LDHA/LDHB which have higher sensitivity than IHC or WB? This could be discussed in discussions.

8. In Figure 3, I observe a variation in the tissue types imaged. Particular, the low expression group from the LDHA, it appears only fat cells have been imaged. Please also comment on the morphological changes ocurring due to the tumor itself. But for IHC comparisons, please try to image similar renal tissue as much as possible.
In the Figure 3 itself, Please label tissue type in each of the three IHC panels.

9. In the supplemental dataset file 1, please mark the lane numbers, the marker being looked at and the molecular weights.
Why was the gel membrane cut?


Minor comments

1. The abstract has sufficient information to stand on its own.
On line 29, ‘however’ would be appropriate rather that ‘whereas'

2. In Introduction
-line 86, Please include which study was the analysis done in?
-line 92 says 'tumor immune?’ This sentence feels incomplete
-Lines 96-99, Please mention if this was in the same set of patients or a separate validation cohort of 150 patients.

3. In Methods, lines 145-146, what was configuration of the microscope, year, model, etc.

4. In Methods, Please specify that the GEPIA2 database uses TCGA datasets. Please also include appropriate citations and links to the publicly available datasets throughout the methods.

5. In Results lines 207-217, please include statistical test next to P values. Authors can also include expression level averages with SD in the same parentheses

6. In Discussions, line 380 Which limited sample size are the authors referring to? Please be specific.

7. In Figure 2B and 2C, Please include all points in bar plot & box and whisker's plot similar to figure 2A

8. In Figure 5, please explain the difference between the labels Tem_CD8 and Tcm_CD8

9. Did the authors consider using Figure 6 data for pathway analysis and for hypothesis generation on the relationship between LDHB, immune cell subtypes and immune inhibitory molecules?

10. In Table 1, Please include what the staining values mean, and how they were obtained in the table footer.
Do the LDHA and LDHB staining values correlate with ccRCC stage. If so, can authors include a figure showing that.

·

Basic reporting

figure 1 needs more clear annotation, "discovery cohort", "validation cohort" need to be more specific with data resources and cohort sizes, the current annotation might confuse the readers to think that the same "discovery" and "validation" cohorts were used across the four tasks (expression profile, prognostic value, immune cells, immune inhibitors).

please check why there are two same plots for “purity” and LDHB expression on Figure 4.

Is the “immunoinhibitors” meaning immune checkpoint inhibitors? Please elaborate the concept. In addition, how did the authors curate the 24 immunoinhibitors, what are the references and is the list comprehensive?

Experimental design

potential overlap of samples across "discovery cohort" and "validation cohort" need to be checked and reported to support the appropriate validation process. It’s essential to use non-overlap discovery and validation cohorts for the validation process. If the databases adapted the same or overlapped samples, the validation would not be reliable.

For the study of “immune cells”, the cell types are not aligned with TIMER2 and TISIDB, how would the authors proceed with the validation process? what conclusion can be drawn if there are no obvious common cell types for these two methods?

Validity of the findings

the authors have shown the differential expression of LDHA/LDHB among tumor and normal samples, however, the conclusions and potential mechanisms of association of LDHA/LDHB with immune infiltration (immune cells and immune inhibitors) are not well explained and described.

the datasets used for discovery and validation of the four tasks (expression, prognostic, immune cells and immune inhibitors) are not well described and can be ambiguous for the readers.

---

## Round 0.2 · Minor Revisions

Please address the remaining concerns of reviewer #2 and amend the manuscript accordingly.

Reviewer 2 ·

Basic reporting

All the comments raised are answered in detail and well explained.

Experimental design

Is fine and satisfactory.

Validity of the findings

The author's response justifies most of the points mentioned.

·

Basic reporting

Authors have improved writing quality and addressed my comments regarding writing, language and addition of references.
Authors have also added additional figures and information in supplemental section to further improve the manuscript.

Experimental design

Authors have addressed most of my comments regarding research methods. Some additional comments are included below.

- Regarding author's response to Major Comment 1:
Please include H&E data in the supplemental, and text in the main manuscript results section explaining the morphological characteristics of the tissue types as assessed by H&E.

- Regarding author's response to Major Comment 9:
Thank you for including this information.
WB MW usually go from higher MW to lower MW, in decreasing order. But in the supplemental data file, the MW markers in the supplemental are reversed (going from 40 Da to 35 kDa. Please explain. 


- Regarding author's response to Minor Comment 2:
Validation cohort is typically a separate group as this makes for a stronger validation.

Validity of the findings

Authors have addressed most of my comments satisfactorily. Some additional comments regarding authors' findings are below -

-Regarding author's response to Major comment 3:
Thank you for the clarification. Please also clarify in the text, wherever possible in parentheses, that Tcm_CD8 are a subset/(memory cell) subtype of CD8+ T cell.

-Regarding author's response to Major Comments 8:
Thank you for including information regarding lipid droplets in ccRCC tissue. This is helpful to the reader and I recommend that authors add this information in the text.

- Regarding author's response to Minor Comments 9:
I recommend authors explore the LDHA & LDHB pathways in Reactome & wiki pathways and do a broader gene ontology (cellular processes, biological functions) assessment. Looking at only immune inhibitory molecules could be limiting as LDHAs connection to immune processes may be indirect or may be cellular trafficking related.

Additional comments

1. Authors have included a discussion paragraph (line 360) to address signaling mechanisms that could reduce LDHB expression.
- Are there public datasets of Renal Cell Carcinoma where authors could look at miRNA data to see if miR-375 is increased?

2. Authors have done a good job responding to minor comment 10. Please include the scoring system used in the methods.

·

Basic reporting

The authors have addressed the reviewer's request of including more detailed information for the discovery and validation cohorts, which would be very helpful for the readers to understand the study design systematically. In addition, the authors have clarified the multiple "purity" plots confusion, and have provided detailed information for "immunoinhibitors".

Experimental design

Thanks for the authors to confirm no overlap among discovery and validation cohorts.

Validity of the findings

Thanks the authors for clarifying the there was no direct interaction between LDHA/LDHB with immune infiltration.

---

## Round 0.3 · accepted · Accept

All remaining issues pointed out by the reviewers were adequately addressed and the manuscript was revised accordingly. Therefore, this version is acceptable now.